# Drenching Bovine Colostrum, Quercetin or Fructo-Oligosaccharides Has No Effect on Health or Survival of Low Birth Weight Piglets

**DOI:** 10.3390/ani12010055

**Published:** 2021-12-28

**Authors:** Kevin Van Tichelen, Sara Prims, Miriam Ayuso, Céline Van Kerschaver, Mario Vandaele, Jeroen Degroote, Steven Van Cruchten, Joris Michiels, Chris Van Ginneken

**Affiliations:** 1Comparative Perinatal Development, Faculty of Biomedical, Pharmaceutical and Veterinary Sciences, Antwerp University, Universiteitsplein 1, 2610 Wilrijk, Belgium; kevin.vantichelen@uantwerpen.be (K.V.T.); sara.prims@uantwerpen.be (S.P.); miriam.ayusohernando@uantwerpen.be (M.A.); steven.vancruchten@uantwerpen.be (S.V.C.); 2Laboratory for Animal Production and Animal Product Quality, Faculty of Bioscience Engineering, Ghent University, Coupure Links 653, 9000 Ghent, Belgium; celine.vankerschaver@ugent.be (C.V.K.); mario.vandaele@ugent.be (M.V.); jerdgroo.degroote@ugent.be (J.D.); joris.michiels@ugent.be (J.M.)

**Keywords:** pig, performance, oral supplementation, neonatal, survival, low birthweight, colostrum, quercetin, scFOS, milk replacer

## Abstract

**Simple Summary:**

Over the past three decades, the litter size of sows has been increased to improve productivity. This has not only led to more piglets per sow, but also an increased proportion of low birth weight piglets, and consequently, higher pre-weaning mortality. In an attempt to improve the resilience of small piglets, an oral supplementation (drenching) with a bioactive compound can be applied. In this study, low birth weight piglets were drenched with bovine colostrum, short-chain fructo-oligosaccharides or quercetin (each dissolved in a plain milk replacer) during the first seven days after birth. The animals’ body weight, mortality, skin lesions, and different blood parameters were evaluated between birth and 2 weeks post-weaning. None of the supplemented compounds had a positive effect on any of the parameters, and thus, on the resilience of low birth weight piglets. Moreover, a negative effect on survival was observed in piglets that were drenched with short-chain fructo-oligosaccharides. These results showed that the evaluated bioactive compounds, in their given dosages, were unable to improve the low birth weight piglets’ survival and emphasized the complex, multifactorial origin of pre-weaning mortality.

**Abstract:**

The introduction of hyperprolific sows has resulted in more low birth weight (LBW) piglets, accompanied by higher mortality. A possible strategy to enhance the resilience and survival of LBW piglets is oral supplementation (drenching) of bioactive substances. This study evaluated the supplementation of bovine colostrum, short-chain fructo-oligosaccharides (scFOS) or quercetin that were dissolved separately in a milk replacer. The study was divided into two sub-experiments. First, the milk replacer was compared with a sham drenched group. Secondly, each dissolved compound was compared with the milk replacer. The LBW piglets, defined as weighing between (mean litter birth weight −1*SD) and (mean litter birth weight −2.5*SD), were randomly allocated to the different treatments and drenched once a day for seven days. On day 1, 3, 9, 24 and 38, piglets were weighed and scored for skin lesions. Blood samples were collected on day 9 and 38 and analyzed to determine glucose, non-esterified fatty acids, urea, immunoglobulin G, insulin-like growth factor 1, and a standard blood panel test. There was no difference between sham drenched piglets and piglets that were drenched with milk replacer regarding any of the parameters. No effect was observed between the milk replacer group and any of the bioactive compounds either, except a higher mortality within the scFOS group. In conclusion, this study showed that drenching the evaluated bioactive compounds, in the used dosages, did not improve LBW piglets’ resilience or survival and more research is required to determine the effect of scFOS on small piglets.

## 1. Introduction

To improve pork production, the modern pig industry has made big investments in the selection of hyperprolific sows, and consequently, in increased litter sizes [1,2,3]. Unfortunately, the increased number of piglets per sow has been accompanied by higher pre-weaning morbidity and mortality and an extended time to reach market slaughter weight due to a proportional increase of piglets with a low birth weight (LBW) [4,5,6,7]. As recently reviewed by Ward et al. [8], the reduced litter uniformity is the result of surpassing uterine capacity and placental insufficiency. Thus, selection towards hyperprolific sows has led to increased economic losses due to higher pre-weaning mortality and has raised animal welfare and public ethical issues for sows and piglets [9,10]. In an attempt to tackle both ethical and economic concerns that are associated with hyperprolific sows and LBW piglets, interventions usually take place at sow, farm, and piglet level [11].

Whereas sow interventions (i.e., providing dietary energy sources or supplements during gestation to improve embryo quality [12,13,14,15,16]) and management strategies (e.g., providing a clean and warm farrowing environment [9,11,17,18] and profound farrowing surveillance [19]) have a more preventive focus, direct interventions at the piglet level are needed as well to ensure better survival chances for LBW piglets. Commonly applied strategies include nurse sows [20], split suckling [20,21], cross-fostering [22,23,24] and artificial rearing [11,25]. Another strategy to enhance the resilience and survival of LBW piglets is through supplementation of energy or bioactive substances [26,27,28,29,30,31,32,33]. Energy supplementation mainly attempts to prevent energy depletion, and consequently, starvation and hypothermia [26,27,32,33]. However, piglets are not only born with low energy reserves but are also immunologically naïve [34]. They require an exogenous source of both energy and immunoglobulins (i.e., colostrum) [8,35]. LBW piglets often fail to retrieve an adequate colostrum intake due to their low viability or their inability to compete with bigger littermates for functional teats [20]. To tackle the reduced colostrum consumption, the supplementation of bovine colostrum has been evaluated in different studies [27,36,37,38,39]. In most of these studies, an improved immunological response was observed after colostrum supplementation [27,37,38]. However, under practical conditions, where small dosages and simple handling are required, the supplementation of colostrum does not always result in better survival. Some authors have found higher survival rates with small dosages of bovine colostrum [27], whereas others could only observe an improved survival when colostrum was supplemented in high volumes [36].

It goes without saying that any form of supplementation requires well-balanced gut health, a trait known to be impaired in LBW piglets [8,40,41,42]. In that respect, providing LBW piglets with prebiotics can be interesting. Short-chain fructo-oligosaccharides (scFOS), prebiotics that consist of non-digestible chains of one glucose molecule linked to two to four fructose molecules, have proven to be beneficial in piglets. scFOS induce a shift in the gut microbiota, favoring *Bifidobacteria* and *Lactobacillus* spp., which efficiently ferment scFOS into short-chain fatty acids [43]. Additionally, scFOS supplementation influences the neonatal immune system by stimulating the secretion of immunoglobulin A, increasing activated T-cells and interferon γ [44,45], and improves performance in terms of body weight at weaning (in piglets with an average birth weight of 1.35 kg) [46]. However, most studies involving scFOS are based on maternal supplementation or supplementation of preterm, weaned, or piglets with a birth weight higher than 1 kg [43,44,45,46,47]. The effect of scFOS on LBW piglets with a birth weight below 1 kg remains unknown.

Another important requirement for a well-functioning digestive tract is the maintenance of an intestinal redox balance. Newborn piglets, and more notably LBW piglets, often suffer from oxidative stress, an imbalance between the production and the ability to eradicate reactive oxygen species (ROS) [48,49,50]. At the gut level, oxidative stress can result in a disrupted intestinal barrier and an abnormal intestinal development that lead to an increased permeability for pathogens or toxins [51,52] and insufficient nutrient absorption [49,53]. In this regard, supplementing LBW piglets with an antioxidant could counterbalance intestinal oxidative stress and its downstream negative effects on gut health. Quercetin, a ubiquitously present flavonoid in plants, is a strong antioxidant [54]. During in vitro studies, quercetin has already proven to have beneficial effects (improved viability, barrier function, and reduced levels of ROS) on H_2_O_2_ stressed intestinal porcine epithelial cells from the jejunum of unsuckled neonatal piglets (IPEC-J2) [55,56,57]. Another study, using IPEC-1 cells, recently showed that quercetin was able to protect the intestinal cells from diquat-induced oxidative damage [58]. However, the translation of quercetin’s positive in vitro results into field trials with LBW piglets remains difficult. No positive effect of quercetin supplementation was found in LBW piglets in terms of body weight and intestinal morphology when given 10 or 50 mg/kg during the first week after birth. However, 10 mg/kg quercetin did appear to improve the intestinal barrier function [59]. Other in vivo studies that examine the effect of quercetin in pigs usually focus on weaned piglets [60,61,62,63,64,65]. Thus, very little is known about the potential effects of quercetin supplementation in neonatal, LBW piglets.

Given the limited knowledge concerning the effects of the previously mentioned compounds on LBW piglets, the current study aimed to examine whether supplementation of bovine colostrum, scFOS, or quercetin during the first week of life, could improve the resilience of LBW piglets. In that respect, each compound was given once daily, during seven days and different zootechnical parameters, i.e., body weight, skin lesions and mortality (on day one (d1), day three (d3), day nine (d9), two days (d24) and two weeks post-weaning (d38)) were examined. Additionally, a blood sample was collected for hematologic and biochemical analysis (on d9 and d38).

## 2. Materials and Methods

### 2.1. Ethical Approval

This study was reviewed and approved by the Ethical Committee for Animal Experimentation of the University of Antwerp (ECD 11/2018) and was compliant with national legislation and European guidelines (2010/63/EC).

### 2.2. Animals

The study was conducted on a commercial farm (Hoogstraten, Belgium). All sows (Topigs20 (*n* = 98), Norwegian Landrace (*n* = 12)) were kept in individual farrowing crates (2.25 × 0.60 m) that were located in pens (2.50 × 1.75 m) with slatted flooring. A nesting area was provided with an unslatted floor and a top cover. The parity of the sows varied from one to ten, with a mean parity of 4.34 ± 2.13 standard deviation (SD). The sows were fed with a commercial gestation diet up to farrowing. After farrowing, all sows were switched to a commercial lactation diet. Piglets included in the study, as well as their littermates, were subjected to the standard handling procedures in the farm: before the age of one week, all piglets were ear-tagged, tail docked, received 200 mg iron dextran I.M. and male piglets were castrated using meloxicam analgesics (0.4 mg/kg I.M.). Piglets were weaned at the age of 22.2 ± 0.6 days.

### 2.3. Piglet Selection

All piglets were weighed within four hours after birth, and subsequently, the mean birth weight of each litter and the SD were calculated. LBW piglets were defined as having a birth weight between (mean birth weight litter-1 SD) and (mean birth weight litter-2.5 SD). A maximum of two LBW piglets was selected in each litter to minimize the effect of sow traits and was identified by a colored ear tag. In total, 188 LBW piglets were selected, spread over six farrowing rounds and 110 sows.

### 2.4. Experimental Treatments

The experimental set-up consisted of two experiments for which the 188 selected piglets were randomly allocated to six treatments: a sham group (*n* = 37), milk replacer (*n* = 38), bovine colostrum (*n* = 38), scFOS (*n* = 39), and quercetin (*n* = 36).

To ensure that the milk replacer, acting as the solvent for the bioactive compounds that were supplemented during the second experiment, did not affect the LBW piglets’ survival or health, a pre-experiment was conducted during which any effect of the milk replacer against the effect of drenching was tested. In this experiment two groups of piglets were included: the sham intervention group (*n* = 37) and the group drenched with a plain milk replacer (*n* = 38). The sham intervention implied a fake drenching by inserting an empty 2.5 mL syringe into the piglet’s mouth for ±20 s. The drenching duration was based on preliminary testing, during which the average catching, and drenching time was 29.6 ± 8.1 s per piglet (average catching time by one person: 10.5 ± 5.9 s; average drenching time: 19.0 ± 5.7 s).

This was repeated once a day during the first week after birth (day one till day seven). The other group of piglets was drenched daily during the first week of life with a plain milk replacer (at 25 °C) which was not enriched with immunoglobulins, prebiotic fibers, or antioxidants (other than propyl gallate or butylated hydroxyanisol) (Table 1). One dose of milk replacer provided 8.95 J metabolizable energy to the drenched LBW piglet.

In a second experiment, the daily oral addition of bioactive compounds to the milk replacer was compared with the daily oral supplementation of a milk replacer. The LBW piglets were randomly allocated to one of four treatments: milk replacer (i.e., same group as during preliminary experiment (*n* = 38)), bovine colostrum (*n* = 38), scFOS (*n* = 39) or quercetin (*n* = 36) supplementation. Within every litter, a maximum of two piglets was selected and allocated to different treatments. Consequently, for every treatment, there was only one piglet per sow.

All groups received a daily oral supplementation of 2 mL at ±25 °C from day one until day seven.

The bovine colostrum (Volostrum^®^, Volac International Ltd., Royston, UK) was dissolved in the aforementioned milk replacer at a concentration of 0.45 g/mL, resulting in supplementation of 0.9 g per piglet. This dosage was chosen, following the results of studies where beneficial aspects of bovine colostrum on growth performance, intestinal development, immune parameters, and sanitary status of pigs were demonstrated when given during the early post-weaning period (dose of 1 mL) [66,67] and pre-weaning period (dose of 1 mL) [68] or as commercially available supplements containing bovine colostrum (dose of 1–2 mL) [27].

Based on previous studies by Le Bourgot et al. [44,69], Apper et al. [47] and Ayuso et al. [46], scFOS (64.8 g/100 mL active product, Profeed L95, Beghin-Meiji, Tereos, Marckolsheim, France) were supplemented to the allocated piglets in a dosage of 1 g scFOS/day (1.54 mL scFOS Profeed L95 + 0.46 mL milk replacer; 2 mL in total).

Quercetin (Sigma-Aldrich, Overijse, Belgium) was supplemented in a volume of 2 mL milk replacer containing 10 mg of quercetin. The dose of 10 mg was based on studies by Cermak et al. [70], Bieger et al. [63], Wein and Wolffram [71], Vergauwen et al. [56], Zou et al. [61] and Van le Thanh et al. [65].

### 2.5. Data Collection

#### 2.5.1. Skin Lesion Scoring

A skin lesion score (for the entire body) was given using the scoring system according to Rundgren and Löfquist [72], Pluske and Williams [73] and Parrat et al. [74]:

0: no lesions

1: <5 superficial lesions (skin unbroken)

2: 5–10 superficial lesions or <5 deep lesions (skin broken and evidence of hemorrhage)

3: >10 superficial lesions or >5 deep lesions

The skin lesion scoring was performed on d1, d3, d9, d24 and d38.

#### 2.5.2. Blood Sampling

On d9 and d38, an 8 mL blood sample was taken from the cranial vena cava. The blood sample was divided into three tubes: one serum tube, one EDTA tube, and one heparin tube. The serum and EDTA tubes were sent to Animal Health Care (Torhout, Belgium) for routine biochemical and hematological analysis. The following biochemical parameters were determined: glucose, non-esterified fatty acids (NEFA), and urea. The hematological analysis determined the levels of red blood cells (RBC), the hematocrit (HCT), the hemoglobin (HGB) levels, the lymphocytes, monocytes, neutrophils, eosinophils, basophils, the total white blood cell (WBC) count, and the platelet (thrombocyte) levels.

The heparin tube was centrifuged at 1500× *g* for 10 min at 4 °C. Next, the plasma was collected and kept at −80 °C until further analysis.

#### 2.5.3. IgG and IGF-1 Analysis

The immunoglobulin G (IgG) and insulin-like growth factor 1 (IGF-1) levels were measured using a porcine competitive inhibition and a sandwich enzyme immunoassay, respectively (IgG: Cloud-Clone Corp., Katy, TX, USA, CEA544Po; IGF-1: Cloud-Clone Corp., Katy, TX, USA, SEA050Po). The collected plasma was diluted (1/2500 and 1/50, respectively) and IgG and IGF-1 levels were determined according to the manufacturer’s instructions. All samples were analyzed in triplicate.

#### 2.5.4. Statistical Analysis

To evaluate the potential effect of drenching on all outcome variables, linear mixed models were fitted in JMP Pro 15.1 (SAS Institute Inc., Cary, NC, USA). Treatment and age were added as fixed effects, and sex was considered a covariate. In addition, all two-way interactions between treatment, age and sex were included. Given the fact that the piglets were selected over ten months (six selection rounds), the farrowing round was added as a random effect. To account for the dependence between littermates and the multiple measurements that were performed on the same piglets, the sow (nested in the farrowing round) and the piglet (nested in sow which was nested in the farrowing round) were included, respectively, as random effects as well. Sows that had been used for piglet selection during previous farrowing rounds were neglected, thus, each sow was only included once. This starting model was simplified using stepwise backwards modelling, during which all non-significant effects were removed from the starting model. Body weight, NEFA, urea, IgG, and IGF-1 levels were log-transformed. The *p*-threshold for significance was set at 0.05. When required, post-hoc analysis with Dunnett’s correction was used to compare different bioactive compounds with the milk replacer (control group). Tukey’s correction was used to compare the different age groups during a post-hoc analysis. Given that not all values had a normal distribution and required a logarithmic transformation, all values are presented as median ± SD. To evaluate the probability of more severe skin lesions occurring in certain treatment or age groups, an ordinal logistic regression model was used. The probability of higher mortality between the different groups was evaluated by Cox’s proportional hazard model. A post-hoc analysis was performed using risk ratios. Additionally, mortality was visualized using Kaplan-Meier curves.

## 3. Results

### 3.1. Milk Replacer Compared with Sham Group

#### 3.1.1. Body Weight

Body weight did not differ between LBW piglets that received milk replacer and piglets that were sham drenched (*p* = 0.808).

#### 3.1.2. Biochemical Analysis

The supplementation of the milk replacer did not affect any of the considered biochemical blood values, i.e., glucose, NEFA, urea, IgG and IGF-1.

#### 3.1.3. Hematological Analysis

Similar to the biochemical analysis, no difference was seen between animals that were sham drenched or received milk replacer for any of the hematological parameters (Table 2).

#### 3.1.4. Skin Lesion Scores

The milk replacer group showed a higher probability of having more severe skin lesions (*p* = 0.018).

#### 3.1.5. Mortality

There was no difference in mortality between LBW piglets that were sham drenched or drenched with the milk replacer.

### 3.2. Bioactive Substances Compared with Milk Replacer

#### 3.2.1. Body Weight

Drenching LBW piglets with bovine colostrum, quercetin or scFOS did not result in a different body weight than LBW piglets that were drenched with milk replacer (*p* = 0.715). Males and females did not have significantly different body weights (*p* = 0.77). As expected, the body weight increased during the experimental period (*p* < 0.001) (Figure 1).

#### 3.2.2. Biochemical Analysis

No treatment or sex effect was observed (Table 3). On day 9, glucose, NEFA, urea and IgG levels were significantly higher than on day 38 (glucose: *p* = 0.009; NEFA: *p* < 0.001; urea: *p* < 0.001; IgG: *p* = 0.029). On day 38, the IGF-1 concentrations were significantly higher than on day 9 (*p* < 0.001) (Table 4).

#### 3.2.3. Hematological Analysis

There were no significant interactions. There was no effect of treatment on any of the blood parameters (Table 3). Sex did not affect any examined hematological parameter. On day 9 blood values were significantly lower than on day 38 for RBCs (*p* < 0.001), HCT (*p* < 0.001), HGB (*p* < 0.001), WBC (*p* < 0.001), lymphocytes (*p* < 0.001), monocytes (*p* < 0.001), neutrophils (*p* = 0.001) and eosinophils (*p* < 0.001). There was no age effect on the basophil levels. Thrombocyte level was significantly higher in LBW piglets at the age of 9 days compared to those at 38 days (*p* = 0.001) (Table 4).

#### 3.2.4. Skin Lesion Scores

The odds of having more skin lesions were not affected by treatment (*p* = 0.248). No sex effect was observed. The probability of observing more severe skin lesions was highest during the post-weaning period: highest on day 24, followed by day 38, 1, 9 and 3 (*p* < 0.001).

#### 3.2.5. Mortality

LBW piglets that were drenched with scFOS were more likely to die than those that were drenched with milk replacer (*p* = 0.034). None of the other treatments affected the mortality rate (Figure 2). No sex effect was observed (*p* = 0.833). The LBW piglets were most likely to die on the day of birth, with the odds of dying decreasing with increasing age (*p* < 0.001).

## 4. Discussion

### 4.1. Effect of Milk Replacer

Given that a plain milk replacer was used as a solvent for all supplemented bioactive substances, a first step consisted of discriminating the possible effect of the milk formula with that of the drenching act on the LBW piglets. Therefore, the supplementation of LBW piglets with the milk replacer during seven days was compared with the sham-drenched group.

Overall, the milk replacer supplementation did not affect the performance or survival of LBW piglets. These results were expected since only a very small dose was given to the animals. Other studies that did find an effect of milk formula supplementation often provided milk in larger dosages (using milk cups or other automated systems) during longer periods or switched to artificially rearing at an older age, as recently reviewed by Huting et al. [75] and Baxter et al. [11].

Similar to the abovementioned performance results, the biochemical and hematological parameters showed no differences between sham drenched LBW piglets and LBW piglets that received the milk replacer.

LBW piglets that were supplemented with the milk replacer were more likely to have severe skin lesions than those that were sham drenched. It was hypothesized in this study that supplementing LBW piglets with health-promoting substances would improve their resilience and vitality and, as a consequence, these piglets could engage more in competitive fights for functional teats. However, given the very small caloric dose of 8.95 kJ per day that was given to the piglets and the lack of any enrichment in the milk formula, the milk replacer was very unlikely to improve their vitality. Moreover, even though skin lesion scoring has been validated to be an indicator of aggressive behavior, the scoring system without registration of injury locations lacks to differentiate between lesions that are the result of reciprocal fights and bullying. Therefore, the higher odds of skin lesions in milk-fed piglets cannot be explained without additional behavioral testing.

### 4.2. Effect of Bioactive Substances

One of the objectives of the current study was to evaluate the effect of bovine colostrum supplementation, in a convenient set-up involving a dosage of 2 mL, during seven days. It was hypothesized that the supplemented colostrum would provide the LBW piglets with milk-borne bioactives, such as immunoglobulins and growth factors, and consequently, improve their resilience, performance, and survival. However, no effect was observed on any of the parameters when colostrum-supplemented LBW piglets were compared to the milk replacer group. These results are in line with an earlier field study by Viehmann and colleagues [68] in which neonatal piglets, including LBW piglets, were supplemented with 1 mL of bovine colostrum during three days. This study did not find any improvement on body weight and cumulative mortality (until ten days after birth) either. However, the authors did observe a prolonged survival time for piglets that were supplemented with colostrum. The animals that died within the first ten days after birth, survived three days longer on average when supplemented with bovine colostrum. In our study, mortality was registered on day 1, 3, 9 and 38, so no exact survival time was calculated. However, at the three different time points (i.e., day 1, 3 and 9) mortality was never significantly different between colostrum and milk replacer supplemented piglets. This indicates that colostrum, supplemented for the first seven days of life, did not enhance the survival time of the LBW piglets during this period. The prolongation of the supplementation period in our study did not improve the results observed by Viehmann et al. [68], possibly because we included piglets with a lower average birth weight than those in their experiment, and consequently, a higher dosage might have been required. Additionally, in our study, and contrarily to Viehmann et al. [68], there was less monitoring during parturition and no heating plates were provided. Perhaps most notably was the absence of any increase in IgG levels in the blood of LBW piglets that were drenched with bovine colostrum. Other studies have observed an increase in IgG or immunological proteins after bovine [27,68] or porcine [76] colostrum supplementation. In our study, IgG levels were below the critical level for pre-weaning survival of 10 mg/mL [77,78] and resembled the IgG levels that were found in a group of piglets that were separated from the sow and fed milk replacer during the critical first 12 h of life [76]. This suggests that the LBW piglets in the present study were unable to ingest enough colostrum. The lower chances to compete against heavier littermates possibly further aggravated the absence of any assistance immediately after birth to achieve their first suckle. Overall, the development of a neonatal colostrum supplementation strategy remains difficult. Some studies find no results after providing piglets with high doses of sow colostrum [76,79], whereas others find positive results on mortality with only small dosages of bovine colostrum [27]. These, often contradicting, results underline the complex, multifactorial cause of pre-weaning mortality. Furthermore, it appears that providing neonatal LBW piglets with a sufficient amount of energy is more important than ensuring that they acquire adequate IgG levels. This is demonstrated by the present study where a low caloric colostrum supplementation failed to improve LBW piglets’ survival, whereas similar studies with high energetic compounds were able to observe a positive effect [11,27,31]. Also, Moreira et al. [36] observed similar IgG levels in piglets that received 120 or 200 mL of porcine colostrum, a higher survival rate in the 200 mL fed group, but no differences in antibody levels in piglets that had died. These results indicated that IgG concentration was most likely not the main factor that influenced the piglets’ survival.

A second bioactive compound, scFOS, was supplemented to examine its effect on the performance and survival of LBW piglets. No effect of scFOS on the body weight, blood parameters, or skin lesions was observed. A study by Schokker et al. [80] did observe an increased body weight at weaning and an altered mucosal gene expression in the gut after fructo-oligosaccharide supplementation. This different result could be attributed to the different supplementation regimen (5× our dose and supplementation for 13 days (days 2 till 14 of age)) and the higher birth weight of the piglets in their study [80]. However, Ayuso et al. [46], supplemented scFOS in the same dosage as in the present study and compared supplementation for 7 days with 21 days. The authors found an increased body weight at weaning and during the post-weaning period in normal birth weight piglets, but not in LBW piglets and piglets with a high birth weight, and only in case the animals were supplemented for 21 days. These results suggest that the absent increase of body weight at weaning in our study was most likely due to the LBW and/or limited drenching period of seven days. In the current study, scFOS supplementation did not only fail to improve LBW piglets’ performance and health but negatively impacted their survival rate. The cumulative mortality was higher in piglets that belonged to the scFOS group, compared to those in the milk replacer group. These results contravene the study by Ayuso et al. [46] where the post-weaning mortality tended to be lower in LBW piglets that were drenched during seven days, albeit not significantly. However, the average birth weight of LBW piglets in the latter study was higher than in our study (1.02 kg vs. 0.86 kg). Our results suggest that birth weight could play an important role in the efficacy of scFOS. It appears that scFOS supplementation longer than 1 week can have positive effects on piglets with a birth weight above 1 kg, but could have detrimental effects in compromised, LBW piglets below 1 kg.

Given that LBW piglets are known to suffer from oxidative stress [48,49,50], quercetin was supplemented to test the hypothesis that the beneficial effects on the oxidative status found in vitro [55,56,57,58] can be translated into an improved resilience of neonatal LBW piglets. However, no treatment effect was observed on any of the measured parameters. These results are consistent with an earlier study by Vergauwen et al. [59] that did not find any effect of quercetin after supplementation of 50 mg quercetin in LBW piglets. The authors selected LBW piglets with similar birth weight and drenched the same dose of quercetin (10 mg/kg) until day 7 after birth, resembling the experimental set-up of the current study. However, Vergauwen et al. did not drench on the first and second day after birth and artificially reared the LBW piglets to ensure a suboptimal redox status. Contrarily, our study consisted of an uninterrupted supplementation during the first seven days after birth and, no piglets were artificially reared. In contrast, other in vivo research on quercetin supplementation in weaned or growing pigs observed inconsistent effects on body weight or growth, although higher dosages up to 25 mg/kg appeared to have a beneficial impact [60,81]. On the other hand, Vergauwen et al. [59] did not observe an improvement in the body weight of LBW piglets after a high-dose supplementation of 50 mg/kg. Thus, more research is needed to understand the impact of quercetin on pig health and performance.

## 5. Conclusions

The present study showed no beneficial effects of any of the supplemented compounds (i.e., bovine colostrum, scFOS, and quercetin) on the zootechnical performance or survival of LBW piglets. The supplementation of scFOS surprisingly showed a negative impact on the LBW piglets’ survival. Thus, more research is required to evaluate the impact of birth weight on the efficacy and possible detrimental effects of scFOS. Moreover, a drenching period of seven days is very labor-intensive. Additionally, it could be interesting to examine not only the dosages but also the combination of different compounds as pre-weaning mortality has a complex, multifactorial origin, as each compound can only tackle one potential underlying cause.

## Figures and Tables

**Figure 1 animals-12-00055-f001:**
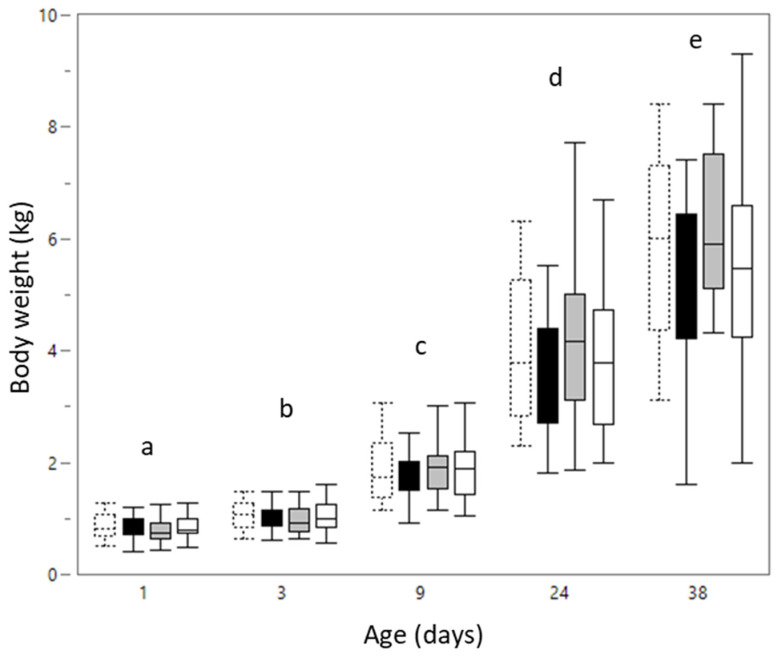
Boxplot of body weight from low birth weight piglets that were drenched with milk replacer (dashed box: d1 *n* = 38, d3 *n* = 30, d9 *n* = 25, d24 *n* = 22, d38 *n* = 21), bovine colostrum (black box: d1 *n* = 38, d3 *n* = 28, d9 *n* = 22, d24 *n* = 19, d38 *n* = 18), short-chain fructo-oligosaccharides (grey box: d1 *n* = 39, d3 *n* = 23, d9 *n* = 15, d24 *n* = 14, d38 *n* = 21 *n* = 11) or quercetin (white box: d1 *n* = 36, d3 *n* = 26, d9 *n* = 21, d24 *n* = 19, d38 *n* = 18) once a day from day 1 until day 7. Body weight is presented at different time points: day 1 (*n* = 151), day 3 (*n* = 107), day 9 (*n* = 83), day 24 (*n* = 74) and day 38 (*n* = 68). Significant age differences (linear mixed models, Tukey post-hoc analysis, *p* ≤ 0.05) are indicated by a different letter (a–e).

**Figure 2 animals-12-00055-f002:**
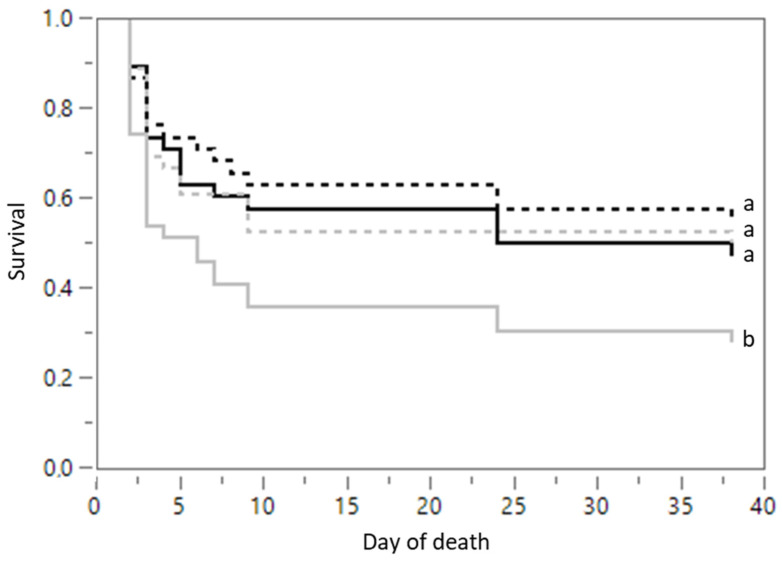
Cumulative mortality of low-birth-weight piglets that were drenched with milk replacer (dashed black line; *n* = 38), colostrum (full black line; *n* = 38), quercetin (dashed grey line; *n* = 36) or short-chain fructo-oligosaccharides (scFOS; full grey line; *n* = 39) over time. Significant differences (Cox’s proportional hazard model, Kaplan-Meier survival plot, *p* ≤ 0.05) between treatments are indicated by a different letter (a–b).

**Table 1 animals-12-00055-t001:** Nutrient, chemical and energetic composition of the supplemented milk replacer.

Analytical Constituents		Nutritional Additives	
Crude protein (%)	19.9	Vitamin A (IU/kg)	25,000
Crude fat (%)	15.9	Vitamin D3 (IU/kg)	5000
Crude ash (%)	7.6	Vitamin E (mg/kg)	80
Crude fibre (%)	0	Vitamin K (mg/kg)	4
Moisture (%)	3.1	Vitamin C (mg/kg)	158
Lactose (%)	38.5	Vitamin B1 (mg/kg)	6
Lysine (%)	1.75	Vitamin B2 (mg/kg)	6
Methionine (%)	0.62	Vitamin B6 (mg/kg)	4
Cystine + Methionine (%)	1	Vitamin B12 (µg/kg)	40
Calcium (%)	0.55	Iodine (mg/kg)	1
Sodium (%)	0.62	Manganese (mg/kg)	45
Phosphorus (%)	0.5	Zinc (mg/kg)	84
Magnesium (%)	0.12	Selenium (mg/kg)	0.30
Iron (mg/kg)	76	Propyl gallate (mg/kg)	3
Copper (mg/kg)	155	Butylated hydroxyanisole (mg/kg)	3
Energetic value			
Metabolizable energy (MJ/kg|kcal/kg)		17.9|4280	
Net energy (MJ/kg|kcal/kg)		14.3|3420	

**Table 2 animals-12-00055-t002:** Blood values (median ± SD) of glucose, non-esterified fatty acids (NEFA), urea, immunoglobulin G (IgG), insulin-like growth factor 1 (IGF-1), red blood cells (RBC), hematocrit (HCT), hemoglobulin (HGB), white blood cells (WBCs), lymphocytes, monocytes, neutrophils, eosinophils, basophils and thrombocytes of low-birth-weight piglets that were supplemented with a milk replacer or sham drenched (linear mixed models, *p* ≤ 0.05).

Dependent Variable	Treatment		
	Milk Replacer	Sham	
	Median ± SD (*n*)	Median ± SD (*n*)	*p*-Value
Glucose (mmol/L)	6.30 ± 0.86 (27)	6.56 ± 1.32 (20)	0.400
NEFA (mmol/L)	0.32 ± 0.56 (29)	0.33 ± 0.82 (19)	0.562
Urea (mmol/L)	2.46 ± 1.37 (27)	2.32 ± 1.40 (20)	0.495
IgG (mg/mL)	5.41 ± 2.81 (12)	2.55 ± 1.79 (12)	0.057
IGF-1 (ng/mL)	27.46 ± 15.31 (12)	20.17 ± 17.62 (12)	0.647
RBC (10^12^/L)	5.29 ± 0.99 (20)	5.46 ± 0.88 (16)	0.151
HCT (%)	33.20 ± 5.51 (20)	33.20 ± 4.15 (16)	0.665
HGB (g/dL)	9.40 ± 1.67 (21)	9.65 ± 1.17 (16)	0.858
WBC (10^3^/µL)	16.30 ± 5.74 (21)	17.81 ± 4.39 (16)	0.606
Lymphocytes (10^3^/µL)	7.17 ± 2.11 (21)	6.93 ± 1.90 (16)	0.960
Monocytes (10^3^/µL)	1.49 ± 0.78 (21)	1.12 ± 0.64 (16)	0.600
Neutrophils (10^3^/µL)	7.23 ± 3.93 (21)	8.40 ± 2.52 (16)	0.984
Eosinophils (10^3^/µL)	0.16 ± 0.12 (21)	0.19 ± 0.17 (16)	0.361
Basophils (10^3^/µL)	0.01 ± 0.01 (21)	0.02 ± 0.01 (16)	0.752
Thrombocytes (10^3^/µL)	404 ± 329.13 (21)	596 ± 334.41 (21)	0.053

**Table 3 animals-12-00055-t003:** Blood values (median ± SD) of glucose, non-esterified fatty acids (NEFA), urea, immunoglobulin G (IgG), insulin-like growth factor 1 (IGF-1), red blood cells (RBC), hematocrit (HCT), hemoglobulin (HGB), white blood cells (WBCs), lymphocytes, monocytes, neutrophils, eosinophils, basophils and thrombocytes, presented by treatment from selected low birth weight piglets (linear mixed models, *p* ≤ 0.05).

Dependent Variable	Treatment				
	Milk Replacer	Colostrum	Quercetin	scFOS	*p*-Value
	Median ± SD (*n*)	Median ± SD (*n*)	Median ± SD (*n*)	Median ± SD (*n*)	
Glucose (mmol/L)	6.30 ± 0.86 (27)	6.00 ± 1.20 (24)	6.17 ± 1.37 (23)	6.23 ± 1.07 (15)	0.466
NEFA (mmol/L)	0.32 ± 0.56 (29)	0.38 ± 0.56 (24)	0.45 ± 0.42 (22)	0.46 ± 0.72 (16)	0.799
Urea (mmol/L)	2.46 ± 1.37 (27)	2.73 ± 1.23 (23)	2.32 ± 1.09 (21)	1.84 ± 1.60 (15)	0.121
IgG (mg/mL)	5.41 ± 2.81 (12)	3.11 ± 1.17 (10)	4.07 ± 3.09 (12)	2.62 ± 2.37 (14)	0.146
IGF-1 (ng/mL)	27.46 ± 15.31 (12)	16.87 ± 15.55 (9)	18.59 ± 13.39 (11)	13.51 ± 17.73 (14)	0.292
RBC (10^12^/L)	5.29 ± 0.99 (20)	5.37 ± 0.96 (15)	5.01 ± 1.11 (18)	5.28 ± 1.07 (10)	0.580
HCT (%)	33.20 ± 5.51 (20)	35.50 ± 3.82 (15)	33.90 ± 4.59 (18)	33.90 ± 4.23 (10)	0.096
HGB (g/dL)	9.40 ± 1.67 (21)	9.80 ± 1.32 (15)	9.05 ± 1.80 (18)	9.85 ± 1.25 (10)	0.151
WBC (10^3^/µL)	16.30 ± 5.74 (21)	19.45 ± 6.39 (15)	16.50 ± 5.27 (18)	15.50 ± 6.07 (10)	0.324
Lymphocytes (10^3^/µL)	7.17 ± 2.11 (21)	8.71 ± 2.10 (15)	6.84 ± 2.23 (18)	6.91 ± 3.31 (10)	0.362
Monocytes (10^3^/µL)	1.49 ± 0.78 (21)	1.06 ± 0.43 (15)	1.39 ± 0.75 (18)	1.18 ± 0.64 (10)	0.295
Neutrophils (10^3^/µL)	7.23 ± 3.93 (21)	8.82 ± 3.30 (14)	8.25 ± 3.47 (18)	8.42 ± 3.02 (10)	0.833
Eosinophils (10^3^/µL)	0.16 ± 0.12 (21)	0.14 ± 0.10 (15)	0.10 ± 0.10 (18)	0.23 ± 0.16 (9)	0.158
Basophils (10^3^/µL)	0.01 ± 0.01 (21)	0.01 ± 0.01 (15)	0.02 ± 0.01 (18)	0.01 ± 0.01 (9)	0.823
Thrombocytes (10^3^/µL)	404 ± 329.13 (21)	546 ± 316.93 (15)	484 ± 335.40 (18)	402 ± 346.78 (10)	0.590

**Table 4 animals-12-00055-t004:** Blood values (median ± SD) of glucose, non-esterified fatty acids (NEFA), urea, immunoglobulin G (IgG), insulin-like growth factor 1 (IGF-1), red blood cells (RBC), hematocrit (HCT), hemoglobulin (HGB), white blood cells (WBCs), lymphocytes, monocytes, neutrophils, eosinophils, basophils and thrombocytes, presented by age and sex from selected low birth weight piglets (linear mixed models, *p* ≤ 0.05).

Dependent Variable	Age			Sex		
	Day 9	Day 38	*p*-Value	Female	Male	*p*-Value
	Median ± SD (*n*)	Median ± SD (*n*)		Median ± SD (*n*)	Median ± SD (*n*)	
Glucose (mmol/L)	6.38 ± 1.14 (54)	5.90 ± 1.07 (35)	0.009	6.17 ± 1.10 (42)	6.18 ± 1.16 (47)	0.466
NEFA (mmol/L)	0.57 ± 0.58 (55)	0.08 ± 0.24 (36)	<0.001	0.43 ± 0.59 (44)	0.39 ± 0.54 (47)	0.974
Urea (mmol/L)	2.65 ± 1.19 (49)	1.76 ± 1.29 (37)	<0.001	2.47 ± 1.29 (40)	2.47 ± 1.34 (46)	0.409
IgG (mg/mL)	4.27 ± 2.66 (24)	2.63 ± 2.37 (24)	0.029	3.91 ± 2.90 (18)	3.22 ± 2.42 (30)	0.843
IGF-1 (ng/mL)	9.19 ± 12.11 (24)	25.35 ± 20.13 (22)	<0.001	20.81 ± 23.12 (18)	15.66 ± 16.17 (28)	0.363
RBC (10^12^/L)	4.27 ± 0.62 (30)	6.09 ± 0.64 (33)	<0.001	5.35 ± 1.03 (32)	5.19 ± 1.03 (31)	0.631
HCT (%)	31.90 ± 4.17 (31)	37.50 ± 3.83 (33)	<0.001	33.55 ± 4.31 (32)	33.90 ± 5.28 (32)	0.359
HGB (g/dL)	8.50 ± 1.26 (31)	10.40 ± 1.07 (33)	<0.001	9.55 ± 1.46 (32)	9.35 ± 1.68 (32)	0.255
WBC (10^3^/µL)	13.51 ± 4.13 (31)	20.38 ± 4.95 (33)	<0.001	16.21 ± 5.66 (32)	17.76 ± 5.98 (32)	0.771
Lymphocytes (10^3^/µL)	5.52 ± 2.00 (31)	8.58 ± 1.70 (33)	<0.001	7.58 ± 2.39 (32)	7.33 ± 2.31 (32)	0.363
Monocytes (10^3^/µL)	0.91 ± 0.33 (31)	1.56 ± 0.72 (33)	<0.001	1.33 ± 0.72 (32)	1.20 ± 0.65 (32)	0.283
Neutrophils (10^3^/µL)	6.85 ± 2.94 (31)	9.42 ± 3.41 (32)	0.001	8.11 ± 3.48 (32)	8.04 ± 3.51 (31)	0.942
Eosinophils (10^3^/µL)	0.08 ± 0.07 (31)	0.21 ± 0.12 (32)	<0.001	0.16 ± 0.12 (31)	0.10 ± 0.12 (32)	0.553
Basophils (10^3^/µL)	0.01 ± 0.01 (31)	0.01 ± 0.01 (32)	0.762	0.02 ± 0.01 (31)	0.01 ± 0.01 (32)	0.064
Thrombocytes (10^3^/µL)	815 ± 367.24 (31)	364 ± 204.30 (33)	0.001	529 ± 330.26 (32)	463 ± 321.85 (32)	0.235

## Data Availability

All data presented in this study is contained within the article.

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
