# Peer review of "Drenching Bovine Colostrum, Quercetin or Fructo-Oligosaccharides Has No Effect on Health or Survival of Low Birth Weight Piglets"

_animals, 2021, doi:10.3390/ani12010055_

Round 1
Reviewer 1 Report
The topic of paper was important for swine production and well written. Here are some suggestions for consideration,
As the results showed there is no effect of bovine colostrum, quercetin and fructo-oligosaccharides on health and survival of low birth weight piglets, maybe the title of the paper can be changed to no effect of bovine colostrum, quercetin and fructo-oligosaccharides on health and survival of low birth weight piglets
Cut down the introduction part, esp., line 62-77
Table 1 & 2 can be present in context or in supplementary material, as they did not provide new information for readers.
Milk replacer compared with sham group must be re-write, or delete. It is no because mile replace does not works, but because the dosage. Piglets consume hundreds ml of milk daily, 2 ml of milk replacer does not make difference, not matter piglet performance or biochemical and haemato-logical parameters, this part can be provided as a pre-research.
Author Response
The authors would first of all like to thank the reviewers for the constructive comments. We have made every effort to include and address all concerns that were pointed out. We hope the improvements to the manuscript are to your liking, but if further clarifications or actions on our part are required, we are at your disposal.
Point-by-point: review comments to the author
Reviewer #1:
- “The topic of paper was important for swine production and well written. Here are some suggestions for consideration.
As the results showed there is no effect of bovine colostrum, quercetin and fructo-oligosaccharides on health and survival of low birth weight piglets, maybe the title of the paper can be changed to no effect of bovine colostrum, quercetin and fructo-oligosaccharides on health and survival of low birth weight piglets”
The authors agree with the reviewer’s opinion regarding the change of title for colostrum and quercetin. However, since a negative effect of fructo-oligosaccharides was observed on mortality, changing the title to ’no effect of bovine colostrum, quercetin and fructo-oligosaccharides on health and survival of low birth weight piglets’ would create wrong expectations for the reader. Therefore, we would like to suggest to the reviewer to keep the present title. Alternatively, and if preferred by the reviewer, the title could be changed to ‘Drenching bovine colostrum, quercetin or fructo-oligosaccharides has no positive effect on health and survival of low birth weight piglets’.
- “Cut down the introduction part, esp., line 62-77”
We agree that the introduction could be cut down, and consequently, emphasize the manuscript’s focus on the interventions at the piglet level. Following the reviewer’s suggestion, the introduction has been cut as followed:
Line 54: ‘Moreover, a significant proportion of LBW piglets suffer from intra-uterine growth retardation for which pigs are known to have a large natural propensity’ has been deleted from the text, since IUGR piglets were not the focus of this study, nor were they discussed any further in the results of discussion section.
Line 62-80: Reduced to 5 lines: ‘Whereas sow interventions (i.e. providing dietary energy sources or supplements during gestation to improve embryo quality) and management strategies (e.g. providing a clean and warm farrowing environment and a profound farrowing surveillance) have a more preventive focus, direct interventions at the piglet level are needed as well to ensure better survival chances for LBW piglets’.
Lines 99-101: The definition of prebiotics has been removed, since this can be considered as otiose information: ‘defined by Gibson et al. as “selectively fermented ingredients that allow specific changes, both in the composition and/or activity in the gastrointestinal microflora that confers benefits upon host wellbeing and health”’
- “Table 1 & 2 can be present in context or in supplementary material, as they did not provide new information for readers.”
The authors agree that table 1 does not provide any additional value to the reader as the sows’ commercial feeding was not the focus of this study and was not incorporated in any of the other sections. Therefore, we have removed the table from the manuscript. However, we would like to keep the table which shows the composition of the milk replacer into the text. This would be an added value for the reader since the section about the milk replacer supplementation has been thoroughly cut, as suggested by the reviewer. This table illustrates that the milk replacer was not enriched.
- “Milk replacer compared with sham group must be re-write, or delete. It is no because mile replace does not works, but because the dosage. Piglets consume hundreds ml of milk daily, 2 ml of milk replacer does not make difference, not matter piglet performance or biochemical and haemato-logical parameters, this part can be provided as a pre-research.”
We agree with the reviewer’s opinion that the section of the milk replacer should be considered as a pre-research. Our intent for this part of the study was never to evaluate whether a 2 mL supplementation of the milk replacer could have beneficial effects for low birth weight piglets, because, as the reviewer correctly mentioned, this dosage would be extremely low and not based on previous literature. Our goal was to ensure that the milk replacer had no effect on any of the parameters that were examined when bioactive substances were drenched, since the milk replacer acted as the solvent for the different compounds. This would allow us to attribute any effect correctly to the compound and not the solvent (milk replacer).
Therefore, we have severely cut the part that discusses the milk replacer to emphasize that this was merely a preliminary experiment.
The figures in the results section that showed the preliminary results of the milk replacer have been removed as these might give the reader the illusion that the milk replacer supplementation was intended to improve the low birth weight piglets’ resilience.
Lines 153-156: The material & methods section emphasizes the pre-research character by the addition of the following text: ‘To ensure that the milk replacer, acting as the solvent for the bioactive compounds that were supplemented during the second experiment, had no effect on the LBW piglets’ survival or health, a pre-experiment was conducted during which any effect of the milk replacer against the effect of drenching was tested.’
Lines 261-268: The results section has been reduced to: ‘Body weight did not differ between LBW piglets that received milk replacer and piglets that were sham drenched (p = 0.808). The supplementation of the milk replacer had no effect on any of the considered biochemical blood values, i.e. glucose NEFA, urea, IgG and IGF-1. Similar to the biochemical analysis, no difference was seen be-tween animals that were sham drenched or received milk replacer for any of the hematological parameters (Table 2). The milk replacer group showed a higher probability of having more severe skin lesions (p = 0.018). There was no difference in mortality between LBW piglets that were sham drenched or drenched with the milk replacer.’
Table 2. Blood values (median ± SD) of glucose, non-esterified fatty acids (NEFA), urea, immunoglobulin G (IgG), insulin-like growth factor 1 (IGF-1), red blood cells (RBC), hematocrit (HCT), hemoglobulin (HGB), white blood cells (WBCs), lymphocytes, monocytes, neutrophils, eosinophils, basophils and thrombocytes of low birth weight piglets that were supplemented with a milk replacer or sham drenched (linear mixed models, p ≤ 0.05).
Dependent variable |
Treatment |
|||
Milk replacer |
Sham |
|
||
|
Median ± SD (n) |
Median ± SD (n) |
p-value |
|
Glucose (mmol/L) |
|
6.30 ± 0.86 (27) |
6.56 ± 1.32 (20) |
0.400 |
NEFA (mmol/L) |
|
0.32 ± 0.56 (29) |
0.33 ± 0.82 (19) |
0.562 |
Urea (mmol/L) |
|
2.46 ± 1.37 (27) |
2.32 ± 1.40 (20) |
0.495 |
IgG (mg/mL) |
|
5.41 ± 2.81 (12) |
2.55 ± 1.79 (12) |
0.057 |
IGF-1 (ng/mL) |
|
27.46 ± 15.31 (12) |
20.17 ± 17.62 (12) |
0.647 |
RBC (1012/L) |
|
5.29 ± 0.99 (20) |
5.46 ± 0.88 (16) |
0.151 |
HCT (%) |
|
33.20 ± 5.51 (20) |
33.20 ± 4.15 (16) |
0.665 |
HGB (g/dL) |
|
9.40 ± 1.67 (21) |
9.65 ± 1.17 (16) |
0.858 |
WBC (103/µL) |
|
16.30 ± 5.74 (21) |
17.81 ± 4.39 (16) |
0.606 |
Lymphocytes (103/µL) |
|
7.17 ± 2.11 (21) |
6.93 ± 1.90 (16) |
0.960 |
Monocytes (103/µL) |
|
1.49 ± 0.78 (21) |
1.12 ± 0.64 (16) |
0.600 |
Neutrophils (103/µL) |
|
7.23 ± 3.93 (21) |
8.40 ± 2.52 (16) |
0.984 |
Eosinophils (103/µL) |
|
0.16 ± 0.12 (21) |
0.19 ± 0.17 (16) |
0.361 |
Basophils (103/µL) |
|
0.01 ± 0.01 (21) |
0.02 ± 0.01 (16) |
0.752 |
Thrombocytes (103/µL) |
|
404 ± 329.13 (21) |
596 ± 334.41 (21) |
0.053 |
Lines 340-364: The discussion section that focusses on the milk replacer has been reduced to the following: ‘Given that a plain milk replacer was used as a solvent for all supplemented bioactive substances, a first step consisted of discriminating the possible effect of the milk formula with that of the drenching act on the LBW piglets. Therefore, the supplementation of LBW piglets with the milk replacer during seven days was compared with the sham-drenched group.
Overall, the milk replacer supplementation did not affect the performance or survival of LBW piglets. These results were expected, since only a very small dose was given to the animals. Other studies that did find an effect of milk formula supplementation often provided milk in larger dosages (using milk cups or other automated systems) during longer periods or switched to artificially rearing at an older age, as recently reviewed by Huting et al. and Baxter et al.
Similar to the abovementioned performance results, the biochemical and haematological parameters showed no differences between sham-drenched LBW piglets and LBW piglets that received the milk replacer.
LBW piglets that were supplemented with the milk replacer were more likely to have severe skin lesions than those that were sham drenched. It was hypothesized in this study that supplementing LBW piglets with health promoting substances would improve their resilience and vitality and, as a consequence, these piglets could engage more in competitive fights for functional teats. However, given the very small caloric dose of 8.95 J per day that was given to the piglets and the lack of any enrichment in the milk formula, the milk replacer was very unlikely to improve their vitality. More-over, even though skin lesion scoring has been validated to be an indicator of aggressive behaviour, the scoring system without registration of injury locations lacks to differentiate between lesions that are the result of reciprocal fights and bullying. There-fore, the higher odds of skin lesions in milk fed piglets cannot be explained without additional behavioural testing.’

Reviewer 2 Report
The study of Tichelen et al. highilghts the effect of of bovine colostrum, quercetin and fructo-oligosac-2 charides on health and survival of low birth weight piglets. The paper is well wirtten, my only concerns are about the experimental design, in discussion session you should justify the use of such a low dose of supplement, as this could be the main reason why you did not seen an effect. In addition, you should specify more clearly the model design since , in the second study, is not clear how many groups and how many piglets x group you had. Also, maybe I missed this point, but it's not clear to me if piglets from the same litter where in the same experimental group or not, because if this was not the case I'm still wondering why you did not used milk cup or other systems to provide larger quantity of the milk replacer.
Minor Comments.
Line 105: Lactobacillus spp.
Line 169: were selected
Line 170 and were
Line 536: Vergauwen et al. there is no date.
Check also the bibliography.
Author Response
The authors would first of all like to thank the reviewers for the constructive comments. We have made every effort to include and address all concerns that were pointed out. We hope the improvements to the manuscript are to your liking, but if further clarifications or actions on our part are required, we are at your disposal.
Point-by-point: review comments to the author
- “The study of Tichelen et al. highilghts the effect of of bovine colostrum, quercetin and fructo-oligosac-2 charides on health and survival of low birth weight piglets. The paper is well wirtten, my only concerns are about the experimental design, in discussion session you should justify the use of such a low dose of supplement, as this could be the main reason why you did not seen an effect.”
We agree with the reviewer’s opinion that the low dosage of supplementation, for the part about the milk replacer, could not have had an effect on the performance of the low birth weight piglets. The section of the milk replacer should be considered as a pre-research. Our intent for this part of the study was never to evaluate whether a 2 mL supplementation of the milk replacer could have beneficial effects for low birth weight piglets, because, as the reviewer correctly mentioned, this dosage would be extremely low and not based on previous literature. Our goal was to ensure that the milk replacer had no effect on any of the parameters that were examined when bioactive substances were drenched, since the milk replacer acted as the solvent for the different compounds.
Therefore, the section about the milk replacer has been severely cut, as suggested by other reviewers as well. If possible, please have a look at our response to reviewer #1.
For the bioactive substances, the dosages were based on existing literature and have been compared with results of other studies during which sometimes positive results were found. We have emphasized or discussed this further in the following sections:
Lines 180-185: ‘This dosage was chosen, following the results of studies where beneficial aspects of bovine colostrum on growth performance, intestinal development, immune parameters and sanitary status of pigs were demonstrated when given during the early post-weaning period (dose of 1 mL) and pre-weaning period (dose of 1 mL) or as commercially available supplements containing bovine colostrum (dose of 1-2 mL).’
Lines 186-189: ‘Based on previous studies by Le Bourgot et al., Apper et al. and Ayuso et al., scFOS (64.8 g/100 mL active product, Profeed L95, Beghin-Meiji, Tereos, Marckolsheim, France) were supplemented to the allocated piglets in a dosage of 1 g scFOS/day (1.54 mL scFOS Profeed L95 + 0.46 mL milk replacer; 2 mL in total).’
Lines 190-193: ‘Quercetin (Sigma-Aldrich, Overijse, Belgium) was supplemented in a volume of 2 mL milk replacer containing 10 mg of quercetin. The dose of 10 mg was based on studies by Cermak et al., Bieger et al., Wein and Wolffram, Vergauwen et al., Zou et al. and Van le Thanh et al..
An additional emphasis on the low dosage has been added:
Lines 383-386: ‘The prolongation of the supplementation period in our study did not improve the results observed by Viehmann et al., possibly because we included piglets with a lower average birth weight than those in their experiment, and consequently, a higher dosage might have been required.’
- “In addition, you should specify more clearly the model design since , in the second study, is not clear how many groups and how many piglets x group you had.”
We would like to thank the reviewer to bring this to our attention. We have clarified the number of animals per group.
Lines 156-158: ‘In this experiment two groups of piglets were included: a sham intervention group (n = 37) vs. a group drenched with a plain milk replacer (n = 38).’
Lines 170-173: ‘The LBW piglets were randomly allocated to one of four treatments: milk replacer (i.e. same group as during preliminary experiment (n = 38)), bovine colostrum (n = 38), scFOS (n = 39) or quercetin (n = 36) supplementation.’
- “Also, maybe I missed this point, but it's not clear to me if piglets from the same litter where in the same experimental group or not, because if this was not the case I'm still wondering why you did not used milk cup or other systems to provide larger quantity of the milk replacer.”
Animals that belonged to the same treatment group were never put together in the same litter (the selected piglets of every sow were only allocated to a maximum of two different treatments, so per sow there were only max. two animals that were drenched and they were always allocated to two different treatments. Therefore milk cup systems could not be used. We have clarified this in the methods section by adding the following text:
Lines 145-146: ‘A maximum of two LBW piglets was selected in each litter to minimize the effect of sow traits and was identified by a coloured ear tag.’
Lines 173-175: ‘Within every litter, a maximum of two piglets was selected and allocated to different treatments. Consequently, for every treatment, there was only one piglet per sow.’
- “Minor Comments:”
Line 105: Lactobacillus spp.
Line 536: Vergauwen et al. there is no date.
Check also the bibliography.
We have checked and corrected/added the mentioned minor comments.

Reviewer 3 Report
This work is hypothesized that supplementing several bioactives will improve the performance of LBW piglets. The outcome was not as anticipated.
This manuscript was far too long; it needed to be more concise and focused on the findings.
Some parts of Materials and Methods were unclear. The authors conducted two sub-experiments, the first with 75 piglets and the second with 151 piglets, but they were only able to select 188 piglets from 110 sows.
The number of piglets used in each group was not mentioned by the authors in the first experiment, but it was in the second experiment as well.
The statistical sections were too long; if possible, show the statistical model used for parameter analysis.
Is it possible to include the weight, number of piglets, skin lesion scores, and mortality rate of each group in the table for both experiments? For example, the number of piglets in the milk replacer group begins (d 1) with 37 piglets and ends with how many piglets remain in this group.
The authors may need to discuss the limitation that dampens the response of piglets to bioactive compounds.
There are far too many references in this manuscript; it is likely that only those references that are most relevant to this work were used.
Author Response
The authors would first of all like to thank the reviewers for the constructive comments. We have made every effort to include and address all concerns that were pointed out. We hope the improvements to the manuscript are to your liking, but if further clarifications or actions on our part are required, we are at your disposal.
“This work is hypothesized that supplementing several bioactives will improve the performance of LBW piglets. The outcome was not as anticipated.”
- “This manuscript was far too long; it needed to be more concise and focused on the findings.”
The authors agree with the reviewer that the manuscript can be improved by focusing more on the findings and shortening the length of the text. We have, following suggestions from other reviewers, reduced the length of the manuscript by four pages, have focused more on the topics that were actually used during the study, cut extremely in the part about milk replacer, have removed unnecessary data, tables, etc. If possible, please have a look as well at our response to reviewer #1.
- “Some parts of Materials and Methods were unclear. The authors conducted two sub-experiments, the first with 75 piglets and the second with 151 piglets, but they were only able to select 188 piglets from 110 sows. The number of piglets used in each group was not mentioned by the authors in the first experiment, but it was in the second experiment as well.”
We would like to thank the reviewer to bring this to our attention. We have clarified the number of animals per group.
Lines 156-158: ‘In this experiment two groups of piglets were included: a sham intervention group (n = 37) vs. a group drenched with a plain milk replacer (n = 38).’
Lines 170-173: ‘The LBW piglets were randomly allocated to one of four treatments: milk replacer (i.e. same group as during preliminary experiment (n = 38)), bovine colostrum (n = 38), scFOS (n = 39) or quercetin (n = 36) supplementation.’
- “The statistical sections were too long; if possible, show the statistical model used for parameter analysis.”
The authors agree that the statistical section could be cut and still allow the reader to understand how the analyses were conducted. We reduced the statistical section in length by removing unnecessary information and removing, as requested by other reviewers, the part about the milk replacer.
Lines 234-255: ‘To evaluate the potential effect of drenching on all outcome variables, linear mixed models were fitted in JMP Pro 15.1 (SAS Institute Inc., Cary, NC, USA). Treatment and age were added as fixed effects and sex was considered a covariate. In addition, all two-way interactions between treatment, age and sex were included. Given the fact that the piglets were selected over a period of ten months (six selection rounds), farrowing round was added as a random effect. To account for the dependence between littermates and the multiple measurements that were performed on the same piglets, the sow (nested in farrowing round) and the piglet (nested in sow which was nested in farrowing round) were included, respectively, as random effects as well. Sows which had been used for piglet selection during previous farrowing rounds were neglected, thus each sow was only included once. This starting model was simplified using stepwise backwards modelling, during which all non-significant effects were removed from the starting model. Body weight, NEFA, urea, IgG, and IGF-1 levels were log transformed. The p-threshold for significance was set at 0.05. When required, post-hoc analysis with Dunnett’s correction was used to compare different bioactive compounds with the milk replacer (control group). Tukey’s correction was used to compare the different age groups during a post-hoc analysis. Given that not all values had a normal distribution and required a logarithmic transformation, all values are presented as median ± SD. To evaluate the probability of more severe skin lesions occurring in certain treatment or age groups, an ordinal logistic regression model was used. The probability of a higher mortality between the different groups was evaluated by a Cox’s proportional hazard model. A post-hoc analysis was performed using risk ratios. Additionally, mortality was visualized using Kaplan-Meier curves.’
- “Is it possible to include the weight, number of piglets, skin lesion scores, and mortality rate of each group in the table for both experiments? For example, the number of piglets in the milk replacer group begins (d 1) with 37 piglets and ends with how many piglets remain in this group.”
We understand the reviewer’s suggestion on expanding the table. The authors have discussed this in the past already, while writing the manuscript, and deliberately did not add all these extra values for the following reasons:
- The tables, which already are quite extensive, would become even more comprehensive and complicated, and consequently, lose focus on the actual blood values and the effect of the treatment, age or sex. The statistical differences are clearly visible in the current presentation. If any additional information were required to explain observed differences, they should be mentioned in the discussion part.
- The mortality rates and body weights were added in the manuscript for the different treatments in the given results sections (figures). Adding them again in the table or text would create duplication of information.
- We have included the number of piglets in the table in every column. This enables the reader to see how many animals/blood samples there were at the different time points (d1, d9, d38), for the different sexes and every treatment.
- “The authors may need to discuss the limitation that dampens the response of piglets to bioactive compounds.”
The authors agree with the reviewer that the manuscript could discuss the limited response better, certainly for the milk replacer. As mentioned earlier, it was never our intent to study a potentially positive effect of the milk replacer on the piglets’ resilience. The first experiment acted as a pre-research to ensure that, if any effect would be found later on with one of the bioactive substances, the effect could by rightfully attributed to the compound and not the milk replacer (solvent). Following the advise of the reviewers, this has been clarified and cut in the manuscript (please see response to reviewers #1 and #2 as well).
To elaborate a bit further on the lack of results for the substances, this could not only be due to the dosage but also to the fact that the piglets were not relocated, and thus, still subjected to competition from larger littermates. This study illustrates that supplementation of the used compounds is subordinate to the competition. If requested by the reviewer, this could be added to the manuscript.
- “There are far too many references in this manuscript; it is likely that only those references that are most relevant to this work were used.”
Following the response to the previous question, by cutting back on the section about milk replacer, the number of references has been reduced by 17. Given the fact that we study three different compounds, the number of references is still high, but necessary to provide the readers with enough background and literature-based comparisons.

Round 2
Reviewer 1 Report
I think the manuscript has been sufficiently improved to warrant publication in Animals, if possible, change the title to Drenching bovine colostrum, quercetin or fructo-oligosaccharides has no positive effect on health and survival of low birth weight piglets.Author Response
Manuscript ID: animals-1496869
Manuscript (previous) title: The effect of bovine colostrum, quercetin and fructo-oligosaccharides on health and survival of low birth weight piglets
The authors would first of all like to thank the reviewers again for extensively reading the manuscript and providing constructive feedback. We have looked into every suggestion and comment and we hope the improvements to the manuscript are to your liking, but if further clarifications or actions on our part are required, we are at your disposal.
The text has also been re-evaluated grammatically and small errors, that have no influence on the content of the text, have been corrected.
Point-by-point: review comments to the author
Reviewer #1:
- “I think the manuscript has been sufficiently improved to warrant publication in Animals, if possible, change the title to Drenching bovine colostrum, quercetin or fructo-oligosaccharides has no positive effect on health and survival of low birth weight piglets.”
We agree to the change of title into: ‘Drenching bovine colostrum, quercetin or fructo-oligosaccharides has no effect on health or survival of low birth weight piglets’.

Reviewer 3 Report
The number of piglets used was not specified. You used 75 piglets in Experiment 1 and 151 piglets in Experiment 2, for a total of 226 piglets. please double-check your declaration of 188 piglets,
Please indicated n of each group at d1, d3, d9, d24 and d38 in Figure 1, as the readers may not understand the number of piglets lost in each treatment.
Please rearrange the following sub-topic of 3.2. Bioactive substances compared with milk replacer as follows: 3.2.1 Body weight, 3.2.2 Mortality, 3.2.3 Skin lesion score and 3.2.4 Biochemical analysis. Because it will make it easier for your readers to follow your results.
Author Response
Manuscript ID: animals-1496869
Manuscript (previous) title: The effect of bovine colostrum, quercetin and fructo-oligosaccharides on health and survival of low birth weight piglets
The authors would first of all like to thank the reviewers again for extensively reading the manuscript and providing constructive feedback. We have looked into every suggestion and comment and we hope the improvements to the manuscript are to your liking, but if further clarifications or actions on our part are required, we are at your disposal.
The text has also been re-evaluated grammatically and small errors, that have no influence on the content of the text, have been corrected.
Point-by-point: review comments to the author
Reviewer #3:
- “The number of piglets used was not specified. You used 75 piglets in Experiment 1 and 151 piglets in Experiment 2, for a total of 226 piglets. please double-check your declaration of 188 piglets.”
We would like to clarify the number of animals that were used:
Sham: n = 37
Milk replacer (this group was used in both sub-experiments): n = 38
Bovine colostrum: n = 38
scFOS: n = 39
Quercetin: n = 36
This brings the total number of piglets to 188 of which 75 (sham + milk replacer) were used during experiment 1 and 151 (milk replacer + bovine colostrum + scFOS + quercetin) were used during experiment 2.
We have clarified this further for the reader in the following lines:
Lines 151-153:’ The experimental set-up consisted of two experiments for which the 188 selected piglets were randomly allocated to six treatments: a sham group (n = 37), milk replacer (n = 38), bovine colostrum (n = 38), scFOS (n = 39) and quercetin (n = 36):’
Experiment 1:
Lines 157-159:‘In this experiment two groups of piglets were included: the sham intervention group (n = 37) the group drenched with a plain milk replacer (n = 38).’
Experiment 2:
Lines 171-174: ‘The LBW piglets were randomly allocated to one of four treatments: milk replacer (i.e. same group as during preliminary experiment (n = 38)), bovine colostrum (n = 38), scFOS (n = 39) or quercetin (n = 36) supplementation.’
- “Please indicated n of each group at d1, d3, d9, d24 and d38 in Figure 1, as the readers may not understand the number of piglets lost in each treatment.”
We have added the number of animals for every treatment group at every time point in the figure’s legend.
Lines 289-297:’Figure 1. Boxplot of body weight from low birth weight piglets that were drenched with milk re-placer (dashed box: d1 n = 38, d3 n = 30, d9 n = 25, d24 n = 22, d38 n = 21), bovine colostrum (black box: d1 n = 38, d3 n = 28, d9 n = 22, d24 n = 19, d38 n = 18), short-chain fructooligosaccharides (grey box: d1 n = 39, d3 n = 23, d9 n = 15, d24 n = 14, d38 n = 21 n = 11) or quercetin (white box: d1 n = 36, d3 n = 26, d9 n = 21, d24 n = 19, d38 n = 18) once a day from day 1 until day 7. Body weight is presented at different time points: day 1 (n = 151), day 3 (n = 107), day 9 (n = 83), day 24 (n = 74) and day 38 (n = 68). Significant age differences (linear mixed models, Tukey post-hoc analysis, p ≤ 0.05) are indicated by a different letter.
- “Please rearrange the following sub-topic of 3.2. Bioactive substances compared with milk replacer as follows: 3.2.1 Body weight, 3.2.2 Mortality, 3.2.3 Skin lesion score and 3.2.4 Biochemical analysis. Because it will make it easier for your readers to follow your results.”
We agree with the reviewer that the suggested structure would be better and have implemented this in the text.
3.1. Milk replacer compared with sham group
3.1.1. Body weight
Body weight did not differ between LBW piglets that received milk replacer and piglets that were sham drenched (p = 0.808).
3.1.2. Biochemical analysis
The supplementation of the milk replacer had no effect on any of the considered biochemical blood values, i.e. glucose NEFA, urea, IgG and IGF-1.
3.1.3. Haematological analysis
Similar to the biochemical analysis, no difference was seen between animals that were sham drenched or received milk replacer for any of the haematological parameters (Table 2).
3.1.4. Skin lesion scores
The milk replacer group showed a higher probability of having more severe skin lesions (p = 0.018).
3.1.5. Mortality
There was no difference in mortality between LBW piglets that were sham drenched or drenched with the milk replacer.
